# Antimicrobial Dispensing Practice in Community Pharmacies in Russia during the COVID-19 Pandemic

**DOI:** 10.3390/antibiotics11050586

**Published:** 2022-04-27

**Authors:** Svetlana Rachina, Roman Kozlov, Anastasiya Kurkova, Ulyana Portnyagina, Shamil Palyutin, Aleksandr Khokhlov, Olga Reshetko, Marina Zhuravleva, Ivan Palagin

**Affiliations:** 1Internal Medicine Department #2, First Moscow State Medical University, 119991 Moscow, Russia; 2Institute of Antimicrobial Chemotherapy, Smolensk State Medical University, 214019 Smolensk, Russia; roman.kozlov@antibiotic.ru (R.K.); anastasiya.kurkova@antibiotic.ru (A.K.); ivan.palagin@gmail.com (I.P.); 3Department of Internal Medicine and General Medical Practice, North-Eastern Federal University, 677007 Yakutsk, Russia; ulyana-nsk@mail.ru; 4Department of Clinical Pharmacology and Drug Ethics, UNESCO, Yaroslavl State Medical University, 150000 Yaroslavl, Russia; shamico@yandex.ru (S.P.); al460935@yandex.ru (A.K.); 5Department of Pharmacology, Saratov State Medical University n.a. V.I. Razumovsky, 410012 Saratov, Russia; reshetko@yandex.ru; 6Department of Pharmacology and Propedeutics of Internal Diseases, First Moscow State Medical University, 119991 Moscow, Russia; mvzhuravleva@mail.ru

**Keywords:** antimicrobials supply, antibiotics, antivirals, COVID-19, community pharmacies

## Abstract

COVID-19 has had a significant impact on health care systems, including drug use. The present study aimed to evaluate the patterns of community supply of antimicrobials from community pharmacies during the COVID-19 pandemic in five cities of Russia. In a cross-sectional study, a random sample of pharmacies reported all episodes of antimicrobials supply during a one-week period. Patterns of supply (age and gender of customer, drug name and formulation, prescription availability, indication, etc.) were analyzed. Altogether, 71 pharmacies took part in the study and 5270 encounters were recorded. In total, 4.2% of visits resulted in supply of more than one antimicrobial agent and 5.2% were for parenteral formulations. The rate of prescription-based purchase in participated cities varied from 40.5 to 99.1%. Systemic antibiotics and antivirals accounted for the majority of supplies (60.5 and 26.3%, respectively). Upper respiratory tract infections were reported as the indication for antimicrobials usage in 36.9% of cases, followed by skin and soft tissue infections (12.1%) and urinary tract infections (8.7%); COVID-19 accounted for 8.4% of all supplies. Amoxicillin with clavulanic acid, azithromycin and amoxicillin were indicated as the top three antimicrobials purchased for upper respiratory tract infections, and azithromycin, umifenovir and levofloxacin were the top three for COVID-19. In general, a high rate of drugs dispensing without prescription was revealed. Antibiotics for systemic use remained the most common antimicrobials, whereas presumably viral upper respiratory tract infections were the main reason for their purchase. COVID-19 infection itself was responsible for a small proportion of the supply of antimicrobial agents, but systemic antibiotics accounted for more than a half of supplies.

## 1. Introduction

Since the beginning of the COVID-19 pandemic, community pharmacists have had an enhanced role in supporting health care systems overloaded by the management of seriously ill patients [1,2,3]. Sick people may have primarily visited pharmacies when seeking professional advice on the management of respiratory symptoms or any other medical condition.

As COVID-19 is a viral infection, a significant increase in sales of antivirals (AV) could have been expected. Nevertheless, the frequent usage of antibiotics (AB) in patients with SARS-CoV-2 infection has been reported across the world [4,5,6]. There are several possible reasons for this. Firstly, some AB (e.g., azithromycin) have been proposed as a part of the treatment algorithms for novel coronavirus infection at the very beginning of the pandemic [7]. Secondly, the perception that COVID-19 is associated with secondary bacterial complications led to the increased demand for AB [8,9]. Finally, it is well known that both health care professionals and the general public still have the bad habit of using AB to treat the symptoms of presumably viral infections, such as the common cold, despite its ineffectiveness [10,11].

This study was undertaken to determine the patterns of community supply of antimicrobials (AM) from community pharmacies during the COVID-19 pandemic in several cities of Russia. Being a part of a larger collaborative project involving members of the WHO Europe Antimicrobial Medicines Consumption Network in non-EU countries of Eastern Europe and Central Asia (“Antimicrobials supplied in community pharmacies in Eastern Europe and Central Asia following the COVID-19 pandemic”), the current study used a common protocol developed by the WHO Regional Office for Europe.

## 2. Methods and Material

This cross-sectional study involved manual data collection from community pharmacies in five cities in various parts of Russia (Moscow, Saratov, Smolensk, Yakutsk and Yaroslavl).

The study was based on methods described in WHO’s “How to investigate drug use in health facilities” that have been widely used by WHO in country profile work and published studies. Sampling methodology was adapted from the World Health Organization & Health Action International 2008 document “Measuring medicine prices, availability, affordability and price components” [12,13].

Due to the population difference, a minimum of 25 pharmacies in Moscow and a minimum of 10 pharmacies in 4 other cities were sought to enable comparisons of patterns of antimicrobial dispensing in different settings.

Pharmacies were chosen by random sampling. If the selected pharmacy declined to participate in the study, another pharmacy was chosen until the required number of pharmacies was recruited. Informed consent was obtained for each participating pharmacy and each pharmacist involved in the study. Every participating pharmacy was allocated a code number known only to the investigators managing the study at the country level. Only the code number (i.e., pharmacy number) was recorded in the data collection form.

A group of pharmacists from participating pharmacies were asked to report all episodes of supply of AM to customers during a one-week period. A minimum of 25 encounters were sought from each participating pharmacy. The provided information included the date of supply, age and gender of the patient/client/customer, the sold AM including its name and formulation (oral, injectable, rectal, etc.). The pharmacists were asked to record whether the request/supply was prescription-based or not. Additionally, they had to report the reason for the supply (presenting symptoms, presumptive or confirmatory diagnosis) in order to allow the analysis of COVID-related or other common infections supplies.

AM were classified according to ATC classification: antibacterials for systemic use (J01), antivirals for systemic use (J05), antimycotics for systemic use (J02), antifungals for dermatological use (D01), antidiarrheals, intestinal antiinflammatory/antiinfective agents (A07) and antiprotozoals (P01).

The relative use of AM from “Access”, “Watch” and “Reserve” groups according to the WHO (AWaRe) classification was evaluated. The 2019 AWaRe list encompasses medicines from the ATC group J01 and several additional agents, namely neomycin (ATC code A07AA01), streptomycin (A07AA04), polymyxin B (A07AA05), kanamycin (A07AA08), vancomycin (A07AA09), colistin (A07AA10), rifamixin (A07AA11), rifampicin (J04AB02), rifamycin (J04AB03), rifabutin (J04AB04) and metronidazole (P01AB01) [14]. AM are assigned to three categories according to the impact of different AB and AB classes on antimicrobial resistance [15]. The “Watch” group includes agents with higher resistance potential, while the “Reserve” agents are suggested as the last-resort AB to treat confirmed or suspected infections caused by multidrug-resistant organisms. The WHO has proposed a country-level target of at least 60% of “Access” group AB out of total AB consumption. In the absence of indication-linked information on AB use, the WHO AWaRe classification allows a more detailed analysis of aggregated data and opportunities for stewardship activities [15].

For the purposes of calculation, each case of pharmaceutical supply is counted here as one prescription, regardless of dosage or duration. Using indication data reported by the pharmacists, the patterns of AM supply for Upper respiratory tract infections (URTI) and COVID-19 were further analyzed. Data were sought from a minimum of 1000 encounters where an AM was supplied. A study coordinator was appointed in each city in order to control the proper collection of data and data entry from all the participating pharmacies in that city. All the data were collected between October 2020 and January 2021. Data were aggregated and presented at the national level and the regional level; the sub-analysis was carried out taking into account the sector of pharmacies (private/public).

The protocol of the study was approved by the WHO Human Research Ethics Committee (N = ERC.0003457) and by the Independent Ethics Committee of Smolensk State Medical University of the Ministry of Health of the Russian Federation (Protocols #228 and #229). Data were analyzed my means of descriptive statistics and were expressed as numbers and percentages.

## 3. Results

### 3.1. Description of Pharmacies

Altogether, 71 pharmacies took part in the study. Most regional pharmacies were part of private pharmacy chains, while all participating pharmacies in Moscow were state-owned public facilities (Table 1). Only one pharmacy in Smolensk was classified as a rural pharmacy.

Information was available for 5270 community pharmacy encounters with one or more AM supplied. There were 1210 encounters reported in Moscow and between 496 and 2150 in the four regions.

### 3.2. Description of Encounters

The age distribution of the customers is shown in Table 2. Out of the 5270 encounters, 60.2% were with female customers, ranging from 57 to 62% depending on the city. Almost half (49.5%) of the customers were aged 36–60 years. Less than 5% encounters related to children and adolescents of 18 years or younger. The proportion of elderly customers (aged > 60 years) varied in regions from 5 to 25.1%.

There were 5514 AM supplied (Table 3). In total, 4.2% of visits resulted in supply of more than one AM agent. Across all ATC codes, 5.2% of supplies were for parenteral formulations. Overall, only 0.4% of encounters resulted in the supply of both oral and parenteral AM agents. The highest rates of supply of parenteral formulations were seen in Smolensk (10.9%).

Overall, 70.5% of encounters involved presentation of a prescription (61.8, 70.1 and 94.5% in private chain, private independent and public pharmacies, respectively). The rate of prescription-based purchase was the lowest in Yakutsk (40.5%) and the highest in Moscow (99.1%). Pharmacists reported the reason for AM use for almost all encounters.

### 3.3. Description of AM Supplied

The most commonly supplied groups of AM belonged to the systemic AB and AV agents, comprising 60.5 and 26.3% of all purchases with available ATC codes, respectively (Table 4). AB for systemic use were the most commonly used in all cities but their proportion varied from 49.6 to 81.1%. The highest rate of supply of AV was recorded in Yaroslavl (38.4%), the lowest was in Moscow (11.9%). Antifungal agents were supplied in 7.2% of encounters in Saratov.

Among AB, macrolides were the most supplied group (14.9%), followed by combinations of penicillins with beta-lactamase inhibitors (such as amoxicillin with clavulanic acid), fluoroquinolones and third-generation cephalosporins, which accounted for 12.3, 11.4 and 7.2% of total supplies, respectively. There were variations in the supply of different classes of AB in children and adults revealed (data shown in the Appendix A). For instance, macrolides and penicillins were used more often in children. Moreover, the proportion of fluoroquinolones increased with age and reached 12.7% of the total supply in patients over 35 years old.

The list of AM agents supplied in this study included 86 different drugs; the top 10 in each region are shown in Table 5. The most commonly supplied agents varied across the cities. In Moscow, 8 of the top 10 supplied agents were J01 AB, whereas in Saratov only 4 of the top 10 agents were AB. Only amoxicillin with clavulanic acid and azithromycin were included in the top 10 agents in all five cities of the study. The AV umifenovir was included in the top 10 agents in four cities, but not in Saratov. The contribution of the top 10 AM to the total supplies accounted for 52.5% (ranging from 45.6 to 78% in regions).

### 3.4. Indication for AM Agents Supplied

Indication for purchase was recorded in 5501 cases. Overall, the URTI were the most common indication for AM usage in 36.9% of cases (Table 6). Flu/influenza was the reported indication in 6.4 to 11% of encounters varying between cities. COVID-19 related reasons for AM agents supply were reported in 8.4% of cases (16.9% of encounters in Yaroslavl). The frequency of other indications of AM supplies, such as skin and soft tissue infections and gastrointestinal infections, varied significantly. Reporting of other not-specified indications was as high as 16.1%.

### 3.5. AM Agents Supplied for URTI

The top 10 AM agents supplied to treat URTI are shown in Table 7. Among all encounters AB accounted up to 72% of supplies. Systemic AB such as amoxicillin plus clavulanic acid and azithromycin were the most commonly used drugs for URTI. The leading AV—pentanedioic acid imidazolyl ethanamide, umifenovir and interferon alfa-2b—together represented 15.7% of supplies. In general, private sector community pharmacies were distinguished by a higher proportion of AV and lower frequency of AB supplies.

### 3.6. AM Agents Supply for COVID-19

There were 462 cases of supply of AM agents that were linked to symptoms or a diagnosis of COVID-19. The most supplied agents varied in public and private sector (Table 8). AB, namely levofloxacin, ceftriaxone and azithromycin, were indicated as the top 3 AM used for the treatment of SARS CoV-2 infection in public in community pharmacies. The private sector was characterized by a higher proportion of AV, especially umifenovir, accounting for 16.1% of the total supplies.

### 3.7. Supply of AM Agents according to the AWaRe Classification

The proportion of supplies of the “Access”, “Watch” and “Reserve” groups depending on location, age and sector is presented in Table 9. In general, the proportion of the “Reserve” group was relatively small across the population. Azithromycin, fluoroquinolones (levofloxacin, ciprofloxacin) and third-generation cephalosporins (cefixime, ceftriaxone) have made the most significant contribution to the “Watch” group AM supplies. The highest rate of the “Watch” group was detected in children less than 5 years old and in Smolensk.

## 4. Discussion

COVID-19 has had a huge impact on health care systems in general and outpatient settings in particular. As for AB use, two opposite trends have been reported during the pandemic. Knight BD et al. demonstrated a reduction in community AB dispensing by 26.5% in Canada for the first 8 months of the COVID-19 compared with the pre-pandemic period [16]. The latest report from the European Centre for Disease Prevention and Control has also shown a decrease in the total AB consumption between 2019 and 2020 in most EU/EEA countries, mainly in primary care [17]. On the contrary, a trend of growing AM consumption in Russia has been reported during the first spike of the COVID-19 pandemic [18]. From January to March of 2020, about 65 million packages of ABs were sold, i.e., sales increased by 13.5% as compared to the same period of the pre-pandemic year. We can speculate that the growth is related to the outbreak of COVID-19. Sulis G. et al. have shown a significant increase in AB sales, particularly azithromycin, during the peak phase of the first COVID-19 epidemic wave in India [19].

This cross-sectional study aimed to evaluate the patterns of supply of AM agents in the community pharmacies of Moscow and four regional cities (Saratov, Smolensk, Yakutsk and Yaroslavl) of different parts of Russia, taking into account the possible impact of the pandemic. When the health care system is overwhelmed, it is expected that pharmacists often become the primary contact when patients seek medical care and, thus, can influence their behavior and patterns of drug usage [1,2,3]. Information about 5270 encounters in 71 drug stores that resulted into AM supply was collected and analyzed. All pharmacies except one were classified as urban; private pharmacies prevailed in the regional cities, whereas only public-sector pharmacies agreed to participate in the study in Moscow. The numbers of encounters with AM supplied included in the study varied from 496 in Yakutsk to 2150 in Saratov. This may relate to the size of pharmacies in the different cities.

There were more female customers in all cities of the study, with the majority of AM purchases made by individuals between the ages of 36–60 years. Almost all reported encounters resulted in supply of one AM agent; overall, 70.5% of encounters involved the presentation of a prescription when acquiring medications. Interestingly, the presentation of a prescription varied from 40.5% of encounters in Yakutsk to 99.1% in Moscow. Perhaps such a high rate of prescription-based purchases in Moscow is due to the higher compliance with the state regulations of AM sales in public pharmacies. Despite enforced regulations in Russia, self-medication and purchase of AB without prescription remains common practice in private drug stores (both chain and independent) even during the COVID-19 pandemic. High rates of self-medication with systemic AB and OTC purchase of them across the country have been reported previously in different studies [10,20,21,22].

The majority of supplied AM were in oral formulation; the supply of parenteral agents was more common among adults over 60 years old (data not shown). Sales of parenteral formulations were slightly higher in Smolensk as compared to the other cities of the study.

Overall, AB for systemic use was the most commonly supplied group of AM agents in all age groups and all regions. The proportion of AV varied from 27.4% to 38.4% of encounters in the regional cities but was lower in Moscow (11.9%). The exact reason for these differences is unknown, but we can assume this might be due to the participation of only state Moscow pharmacies in the study. All systemic ABs are prescription drugs, while, on the contrary, most AVs, such as umifenovir, belong to the OTC group. It is possible that a patient with a prescription prone to go to the state pharmacy to buy ABs, and on the contrary one will prefer a private pharmacy when AV drugs are needed.

Among Abs, the highest overall rate was seen for macrolides, followed by combinations of penicillins with beta-lactamase inhibitors, such as amoxicillin with clavulanic acid, and fluoroquinolones. Some features have been identified in the supply of AM agents inside different age groups. As an example, it has been shown that macrolides and penicillins were more commonly purchased for children. Along with this, the proportion of fluoroquinolones reached the highest level among patients over 35 years old. These peculiarities can be associated with age restrictions for the use of certain AB classes, such as quinolones and tetracyclines in children. In adults over 60 years old, increased proportion of parenteral third-generation cephalosporins and quinolones and decreased rate of macrolides and penicillins usage was seen. It can be speculated that the difference in use of AM agents in different age groups is due to various reasons, as well as there being a difference in the etiology of infections associated with age and presence of certain comorbidities. Thus, age and comorbidities have an impact on the prescription of AB for outpatient treatment of community-acquired pneumonia in the guidelines [23,24,25]. Another reason is the assumption of a possible age-related variability in the absorption of oral medications and compliance to oral treatment. In this case, elderly patients and the youngest children become the most vulnerable.

Among the most commonly prescribed AM agents the top 10 agents accounted for more than 52% of supplies. Interestingly, the list of most commonly purchased drugs varied across cities. Thus, only two drugs—amoxicillin with clavulanic acid and azithromycin—were in the top 10 for all five cities of the study. It is possible that the discrepancies relate to different indications for the purchase of AM; however, we cannot also exclude the variable practice of AB usage for the same indication, which has been demonstrated in a previous study [11].

It was unsurprising that URTI were the most commonly named indications for the supply of AM agents in community pharmacies (recorded in 37% of cases). Together with COVID-19 and influenza, the overall frequency for purchasing AM agents for “respiratory tract problems” accounted for 53.6% of drug supplies. These data are generally in line with those found in similar studies. Belkina T. et al. evaluated the attitudes of community pharmacists regarding AB use and self-medication in the Saint-Petersburg and Leningrad region of Russia [22]. ABs were mostly used to self-treat upper and low respiratory tract infections (53.3% and 19.3%, respectively); other conditions were dental problems and urogenital infections.

Systemic ABs, namely amoxicillin with clavulanic acid, azithromycin, amoxicillin and AV pentanedioic acid imidazolyl ethanamide (known in Russia and some former USSR countries as ingavirin), were the leading agents for URTI treatment. The latter is used mostly for influenza and common cold of viral etiology in adults and children over 3 years old. In general, the most commonly prescribed ABs corresponded to the included in the national clinical guidelines for the treatment of community-acquired respiratory tract infections [23,26,27]. The appropriateness of prescribing ABs itself cannot be assessed due to the study design. Nevertheless, it should be emphasized that most acute URTIs are caused by viruses and, in principle, do not require systemic AB therapy. Moreover, it is worth mentioning that public pharmacies were characterized by much higher proportion of AB purchases and limited supplies of AVs. We can assume that since prescription-based purchases were much more likely in public pharmacies, patients with bacterial URTIs, where the prescription of ABs is justified, could predominate among them.

COVID-19 accounted for 8.4% of AM supplies. There were a variety of both AV and AB agents used; the top five in the ranking were azithromycin, umifenovir, levofloxacin, amoxicillin with clavulanic acid and ceftriaxone. Our results are consistent with a review published by Chedid M et al., where fluoroquinolones, ceftriaxone and azithromycin were the most frequently prescribed ABs in patients with confirmed SARS-CoV-2 infection [28]. It should be emphasized that the approach towards AB usage has changed after gaining new knowledge and experience in the treatment of novel coronavirus infection. Early studies on the presence of antiviral activity in some AMs, such as azithromycin and hydroxychloroquine, were not confirmed by the results of subsequent more robust studies [29]. According to the available reports, a bacterial and fungal coinfection in patients presenting with the COVID-19 appears to be low and much less than in the previous influenza pandemics [30,31]. Thus, ABs should not be routinely prescribed in case of confirmed SARS-CoV-2 infection. This statement is in line with national guidelines for COVID-19 treatment updated on a regular basis [32].

When analyzing the choice of antiviral drugs, it is necessary to take into account the fact that the data were collected before the launch of new AVs on the Russian market, such as remdesivir, molnupirovir and sotrovimab, which are now widely used for outpatients with SARS-CoV-2 infections across the world.

In order to assist in the development of tools for AM stewardship at local, national and global levels and to reduce antimicrobial resistance, the AWaRe classification of AM agents was proposed by WHO [14]. According to this classification, ABs are classified into different groups to emphasize the importance of their appropriate use. Our data show that the “Watch” group of AM agents reached, in our study, about 60% of supplies, conflicting recommendations that 60% of all consumed ABs must come from the “Access” group. A similar proportion was seen in the public and private sector, among different age groups, with a peak rate in the youngest children (68.2%). It is worth mentioning that the “Access” group agents supplies prevailed during the study period in Yakutsk only. Other cities, including Moscow, were in line with the general trend. Unfortunately, a high proportion of “Watch” and “Reserve” ABs was also seen in Russian hospital settings [33]. This emphasizes the need for a global review of antimicrobial stewardship and approaches towards AM usage aiming to contain antimicrobial resistance.

## 5. Strength and Limitations

We recognize our study has some limitations. The relatively small number of sites means that there is a potential sample selection bias and this will limit the extrapolation of findings to the whole country. Participation in the study was voluntary, and therefore, it is possible that the most conscientious pharmacists and those complying with the legislation agreed to participate in the research. In Moscow, for not very clear reasons, commercial pharmacies refused to participate in the study and the sample was formed only from public sector pharmacies.

The research was based on the self-reported data, so validity cannot be guaranteed. However, reports of practicing pharmacists and a standard data collection tool should maximize the collection of valid and reliable data. Local study coordinators from the research group engaged to conduct the study undertook regular supervision of the pharmacists and reviewed the adequacy of the data collection.

Data on indication were based on pharmacists’ or customers’ assessment and were not verified by the review of prescriptions or medical records. This, to some extent, limited our ability to assess the appropriateness of the prescription and rationality of the choice of AM agents. In addition to this, ‘other indications’ accounted for 16% of supplies. Nevertheless, it demonstrates the patterns of AM supply in community pharmacies and the main reasons why patients present to the pharmacy for the purchase of AM agents, including self-medication. It also makes possible to assess the real practice of AB choice in the community and allows to make some comparisons with the national guidelines. From this point of view, the study may help to identify problematic supply and potential targets for pharmacy-based interventions to improve the community use of AM agents.

## 6. Conclusions

The study evaluated the practice of dispensing AM agents by community pharmacies, and thus, highlighted the key patterns of supply during the first year of the COVID-19 pandemic in Russia. In the context of limited electronic medical records and high prevalence of self-medication, this tool helps to understand how AM agents are used in real practice among outpatients.

In general, a high rate of drug dispensing with no prescription by private pharmacies and unacceptably high proportion of “Watch” and “Reserve” groups of AB was revealed in most cities of the study. It is important to note that AB for systemic use remain the most common AM agents, whereas URTI are the main indication for their purchase. COVID-19 infection itself was responsible for small proportion of AM agents supply, but systemic AB accounted for more than half of the supplies, with azithromycin being the leading AM agent for this indication.

## Figures and Tables

**Table 1 antibiotics-11-00586-t001:** Main characteristics of community pharmacies.

Location	Number of Pharmacies	Sector			Urban/ Rural Setting	Number of Encounters
		Private Chain	Private Independent	Public		
Moscow	26	0	0	26	26/0	1210
Saratov	13	13	0	0	13/0	2150
Yaroslavl	12	10	1	1	12/0	790
Smolensk	10	9	1	0	9/1	625
Yakutsk	10	5	2	3	10/0	496
Total	71	37	4	30	70/1	5270

**Table 2 antibiotics-11-00586-t002:** Gender and age distribution of AM encounters in community pharmacies.

	Moscow	Saratov	Yaroslavl	Smolensk	Yakutsk	Total
Number of encounters	1210	2149	790	625	497	5270
Female, %	57.0	61.0	62.0	61.0	61.0	60.2
**Age distribution**						
<5 years, %	2.0	0	2.0	2.0	1.0	1.1
5–12 years, %	2.2	0.3	2.4	3.5	3.2	1.7
13–18 years, %	1.2	0.7	1.6	2.1	2.6	1.3
19–35 years, %	24.0	28	28	32.0	39.0	28.5
36–60 years, %	56.6	46.2	53.9	42.6	48.6	49.5
>60 years, %	13.7	25.1	12.5	18.2	5.0	17.9

**Table 3 antibiotics-11-00586-t003:** Description of AM encounters in community pharmacies.

	Moscow	Saratov	Yaroslavl	Smolensk	Yakutsk	Total
Number of AM supplied	1253	2199	856	668	538	5514
% of encounters with >1 AM supplied	3.0	2.3	7.3	6.2	7.5	4.2
% of encounters with parenteral AM	7.1	3.7	2.5	10.9	2.5	5.2
% of encounters with oral and parenteral AM	0.5	0.1	0	1.9	0.2	0.4
% of encounters with reported reason for AM use	100	99.9	100	100	100	99.9
% of encounters with prescription	99.1	60.0	76.2	68.4	40.5	70.5
% of encounters with emergency use *	0.7	0	2.1	0.1	5.6	1.0
% of encounters with other reasons to use **	0.2	40.0	21.7	31.3	53.9	30.5

Comments: * Emergency supply is defined by law, e.g., in case of emergency needs during the time doctors’ consultation is not available; ** upon request, an oral recommendation from a doctor and/or pharmacist, for prophylaxis or self-medication.

**Table 4 antibiotics-11-00586-t004:** Groups of AM agents supplied in community pharmacies.

Group of AM (ATC Code)	Moscow	Saratov	Yaroslavl	Smolensk	Yakutsk	Total
Antibacterials for systemic use (J01), %	81.1	49.6	56.1	59.6	54.6	60.5
Antivirals for systemic use (J05), %	11.9	27.4	38.4	32.8	32.5	26.3
Antimycotics for systemic use (J02), %	2.9	8.8	4.1	3.6	7.1	5.8
Antifungals for derma-tological use (D01), %	0.6	7.2	0.7	2.0	1.8	3.3
Antidiarrheals, intestinal antiinflammatory/anti-infective agents (A07), %	1.0	4.3	0.1	1.3	1.6	2.2
Antiprotozoals (P01), %	2.5	2.7	0.6	0.7	2.4	2.0
Total number *	1247	1754	713	612	507	4833

Comments: * Supply of AM agents with available ATC codes is presented.

**Table 5 antibiotics-11-00586-t005:** Top 10 AM agents supplied in community pharmacies.

AM Name	% of Total AM Supplied (Rank by Volume)					
	Moscow	Saratov	Yaroslavl	Smolensk	Yakutsk	Total
Amoxicillin + clavulanic acid (AB)	18.7 (1)	7.9 (1)	6.9 (5)	5.7 (6)	12.6 (2)	10.4 (1)
Azithromycin (AB)	16.6 (2)	3.3 (8)	14.4 (1)	12.1 (1)	7.2 (5)	9.5 (2)
Umifenovir (AV)	3.0 (9)		8.3 (4)	7.8 (3)	21.9 (1)	5.5 (3)
Pentanedioic acid imidazolyl ethanamide (AV)		4.2 (4)	13.2 (2)	10.3 (2)		5.3 (4)
Fluconazole (AM)		5.9 (3)	3.3 (9)		6.7 (6)	4.6 (5)
Amoxicillin (AB)	3.3 (7)		6.5 (6)		8.5 (4)	4.1 (6)
Levofloxacin (AB)	8.9 (3)		5.4 (7)	6.6 (5)	2.8 (8)	3.9 (7)
Ciprofloxacin (AB)	6.0 (5)		3.0 (10)	3.6 (10)	8.7 (3)	3.4 (8)
Cefixime (AB)	3.2 (8)	3.8 (6)				3.0 (9)
Ceftriaxone (AB)	6.2 (4)			7.3 (4)		2.8 (10)
Clarithromycin (AB)	3.4 (6)					
Fluconazole + doxycycline (AM + AB)	2.9 (10)					
Acyclovir (AV)		7.0 (2)		3.7 (9)	2.2 (10)	
Interferon alfa-2b (AV)		4.2 (5)	11.3 (3)			
Tilorone (AV)		3.4 (7)	4.9 (8)			
Metronidazole (AP)		3.1 (9)		5.4 (7)		
Rimantadine (AV)		2.8 (10)		4.0 (8)	5.0 (7)	
Tetracycline (AB)					2.4 (9)	

Comments: AB—antibiotic, AV—antiviral agent, AM—antimycotic, AP—antiprotozoal agent.

**Table 6 antibiotics-11-00586-t006:** Recorded indications for AM supplied in community pharmacies, %.

Indication	Moscow	Saratov	Yaroslavl	Smolensk	Yakutsk	Total
Upper respiratory tract infections	46.4	30.3	41.8	35.3	34.9	36.9
Skin and soft tissue infections	5.7	20.4	6.2	8.4	1.3	12.1
Urinary tract infections	11.0	9.2	6.0	6.1	8.4	8.7
COVID-19	6.2	4.6	16.9	12.3	10.4	8.4
Flu/influenza	8.9	6.4	10.5	8.1	11.0	8.3
Gastrointestinal infections	4.7	8.5	0.8	2.4	2.4	5.1
Eye infection	1.4	6.2	2.6	3.1	4.4	4.0
Hospital treatment related	0.2	0.8	0	1.2	0	0.5
Other indications	15.5	13.4	15.2	23.0	21.6	16.1

**Table 7 antibiotics-11-00586-t007:** Top 10 AM agents supplied for URTI in community pharmacies.

AM Agent	% of Total AM Supplied (Rank by Volume)		
	Public Sector	Private Sector	Total
Amoxicillin + clavulanic acid (AB)	26.3 (1)	14.4 (1)	18.0 (1)
Azithromycin (AB)	26.0 (2)	12.9 (2)	16.9 (2)
Amoxicillin (AB)	5.0 (5)	7.7 (4)	6.9 (3)
Pentanedioic acid imidazolyl ethanamide (AV)		9.6 (3)	6.7 (4)
Cefixime (AB)	4.2 (7)	7.0 (5)	6.2 (5)
Levofloxacin (AB)	11.2 (3)	3.6 (9)	5.9 (6)
Umifenovir (AV)	1.5 (10)	6.7 (6)	5.1 (7)
Ceftriaxone (AB)	8.1 (4)	3.3 (10)	4.8 (8)
Interferon alfa-2b (AV)		5.5 (7)	3.9 (9)
Ciprofloxacin (AB)	4.5 (6)		3.4 (10)
Doxycycline (AB)	2.6 (8)		
Josamycine (AB)	1.6 (9)		
Tilorone (AV)		4.0 (8)	

Comments: AB—antibiotic, AV—antiviral agent.

**Table 8 antibiotics-11-00586-t008:** Top 10 AM supplied for the treatment of COVID-19 in community pharmacies.

AM Agent	% of Total AM Supplied (Rank by Volume)		
	Public Sector	Private Sector	Total
Azithromycin (AB)	13.4 (3)	20.5 (1)	19.3 (1)
Umifenovir (AV)	6.1 (6)	16.1 (2)	14.3 (2)
Levofloxacin (AB)	23.2 (1)	10.0 (3)	12.3 (3)
Amoxicillin + clavulanic acid (AB)	11.0 (4)	9.7 (4)	10.0 (4)
Ceftriaxone (AB)	20.7 (2)	4.2 (8)	7.1 (5)
Interferon alfa-2b (AV)		7.4 (5)	6.3 (6)
Pentanedioic acid imidazolyl ethanamide (AV)		6.8 (6)	5.6 (7)
Favipiravir (AV)	4.9 (8)	3.4 (9)	3.7 (8)
Amoxicillin (AB)		4.2 (7)	3.5 (9)
Oseltamivir (AV)	4.9 (9)	2.9 (10)	3.2 (10)
Clarithromycin (AB)	6.1 (5)		
Hydroxychloroquine (AP)	4.9 (7)		
Famciclovir (AV)	2.4 (10)		

Comments: AB—antibiotic, AV—antiviral agent, AP—antiprotozoal agent.

**Table 9 antibiotics-11-00586-t009:** Supplies according to the AWaRe classification, % of total supplies.

	Access	Watch	Reserve
**Location**			
Moscow	36.9%	63.1%	0.8%
Saratov	43.4%	56.6%	4.0%
Yaroslavl	33.2%	66.8%	1.0%
Smolensk	26.2%	73.8%	1.1%
Yakutsk	50.7%	49.3%	1.4%
**Age**			
<5 years of age	31.8%	68.2%	0.0%
5–12 years	47.8%	52.2%	0.0%
13–18 years	48.7%	51.3%	0.0%
19–35 years	41.2%	58.8%	2.3%
36–60 years	36.6%	63.4%	2.3%
>60 years	38.3%	61.7%	0.8%
**Sector**			
Private pharmacies	38.4%	61.6%	2.5%
Public pharmacies	38.0%	62.0%	1.0%

## Data Availability

The data that support the findings of this study are available from the corresponding author (S.R.) on special request.

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
