# Peer review of "Antimicrobial Dispensing Practice in Community Pharmacies in Russia during the COVID-19 Pandemic"

_antibiotics, 2022, doi:10.3390/antibiotics11050586_

Round 1

Reviewer 1 Report

The manuscript "Antimicrobial dispensing practice in community pharmacies in Russia during COVID-19 pandemic" gave us excellent inside in the antibiotic dispensing in 71 pharmaciest in several cities in Russia during the one week of the Covid19 pandemic. The research is well designed and protocol was developed by WHO Regional Office for Europe. The results are well presented and the conclusion is clear and can be useful for the future strategy in antibiotic politics in Russia. The importance of research is to understanding the global situation on the (over)use of antibiotics in the world during the COVID19 pandemic and therefore I suggest that this research should be published in the journal Antibiotics.    

Author Response

Reviewer 1

The manuscript "Antimicrobial dispensing practice in community pharmacies in Russia during COVID-19 pandemic" gave us excellent inside in the antibiotic dispensing in 71 pharmacies in several cities in Russia during the one week of the Covid19 pandemic. The research is well designed and protocol was developed by WHO Regional Office for Europe. The results are well presented and the conclusion is clear and can be useful for the future strategy in antibiotic politics in Russia. The importance of research is to understanding the global situation on the (over)use of antibiotics in the world during the COVID19 pandemic and therefore I suggest that this research should be published in the journal Antibiotics. 

- Thank you very much for your most stimulating appreciation of our work.

Reviewer 2 Report

In general, the paper was well-written.

It adequately highlighted the over-dispensing of antimicrobials by pharmacies, which is actually rampant across the globe and not just in Russia. Particularly alarming was the high (60%) rate of dispensing without a prescription in Yakutsk - I hope the local authorities will look into this.

I find that in certain regions (e.g. Smolensk), quite alot of parenteral formulations were dispensed. Why would community pharmacies sell (or customers buy) antimicrobials which require IV/IM administration?

Curiously, there were no data on ivermectin and hydroxychloroquine sales in Russia. These two AMs received much attention during the initial waves of the COVID-19 pandemic.

Author Response

Reviewer 2

In general, the paper was well-written.

- First of all, thank you for your fair review.

It adequately highlighted the over-dispensing of antimicrobials by pharmacies, which is actually rampant across the globe and not just in Russia. Particularly alarming was the high (60%) rate of dispensing without a prescription in Yakutsk - I hope the local authorities will look into this.

I find that in certain regions (e.g. Smolensk), quite a lot of parenteral formulations were dispensed. Why would community pharmacies sell (or customers buy) antimicrobials which require IV/IM administration?

-  Indeed, in some regions of the Russian Federation, in particular in Smolensk, a high proportion of parenteral ABs was dispensed. Perhaps this is an "echo" of the practice that existed in the USSR, where the availability of oral forms of antibiotics with high bioavailability was limited. A high proportion of parenteral prescriptions, for example, in the outpatient treatment of community-acquired pneumonia in children and adults, was identified in our previous studies (https://cmac-journal.ru/en/publication/2000/3/cmac-2000-t02-n3-p074/). A recent survey of the population showed that patients often copy the practice of doctors. In particular, parenteral ceftriaxone was mentioned by patients who bought antibiotics without a prescription (Front Pharmacol 2022 Jan 31;13:800695. doi: 10.3389/fphar.2022.800695). 

Curiously, there were no data on ivermectin and hydroxychloroquine sales in Russia. These two AMs received much attention during the initial waves of the COVID-19 pandemic.

- Ivermectin is available in Russia only in topical formulations and it was not recommended for COVID-19 treatment in Russian national guidelines. As for hydroxychloroquine it was supplied, mostly in public pharmacies (ranked 7/10 in Table 8 of the paper).

Reviewer 3 Report

The present manuscript "Antimicrobial dispensing practice in community pharmacies in Russia during COVID-19 pandemic",  is a work that highlights the intense problem of prescribing a large number of antibiotics even in cases of viral infections such as COVID 19. 

It is an interesting study but with statistical limitations such as the number of pharmacies selected in the Russian territory and that is a research based on the self-reported data. These are points that the authors comment in the manuscript. 

However, in my opinion the authors can add some of the information stated as limitations (if this is possible) to improve the "strenght" of their research.

Author Response

Reviewer 3

The present manuscript "Antimicrobial dispensing practice in community pharmacies in Russia during COVID-19 pandemic", is a work that highlights the intense problem of prescribing a large number of antibiotics even in cases of viral infections such as COVID 19. 

It is an interesting study but with statistical limitations such as the number of pharmacies selected in the Russian territory and that is a research based on the self-reported data. These are points that the authors comment in the manuscript. 

However, in my opinion the authors can add some of the information stated as limitations (if this is possible) to improve the "strength" of their research.

- Thank you very much for your evaluation and fair comments. We acknowledge all the limitations of the research and tried to be as precise as possible mentioning them. Perhaps, it should be emphasized that our project was part of international one, and we followed the protocol proposed by WHO experts.

Reviewer 4 Report

Dear Author,

Thanks for submitting the manuscript to Antibiotics. It is a well designed and well executed study.

Please include the following:

1) Apply relevant statistical tests.

2) How do you minimize bias? How do you plan for your confounders? Please elaborate.

3) Please state the strengths and limitations of your study?

4) What is the novelty of your work? Several types of studies have been done previously globally? How do you differentiate your work from others?

5) Literature review and search not proper. Many relevant references are missing. 

Author Response

Reviewer 4

Thanks for submitting the manuscript to Antibiotics. It is a well designed and well executed study.

- Many thanks!

Please include the following:

1) Apply relevant statistical tests.

- As the research was a descriptive one and there was no hypothesis tested the results were analyzed by means of descriptive statistics only (frequencies and percentages). We’ve expanded this Part in “Materials and Methods” section.

2) How do you minimize bias? How do you plan for your confounders? Please elaborate.

- The study was based on methods described in WHO’s “How to investigate drug use in health facilities” that have been widely used by WHO in published studies. The sampling methodology was adapted from the World Health Organization & Health Action International 2008 document “Measuring medicine prices, availability, affordability and price components”. So, we’ve used a standard data collection tool (protocol, registration forms) to minimize biases and maximize the collection of valid and reliable data. Local study coordinators undertook regular supervision of the pharmacists and reviewed the adequacy of the data collection. We’ve expanded this Part in “Materials and Methods” section.

3) Please state the strengths and limitations of your study?

- Limitations of the study:

The relatively small number of sites will limit extrapolation of findings to the whole country. Participation in the study was voluntary and therefore, it is possible that the most conscientious pharmacists and those complying with the legislation agreed to participate in the research. The research was based on the self-reported data. Data on indication was based on pharmacists or customers assessment and were not verified by the review of prescriptions or medical records; this, to some extent, limited our ability to assess the appropriateness of the prescription and rationality of the choice of AM agents.

- Strength:

The research was based on the methods described in WHO recommendations to conduct such a study:

  1. How to investigate drug use in health facilities: selected drug use indicators. Available at: https://apps.who.int/iris/handle/10665/60519.
  2. Measuring medicine prices, availability, affordability and price components. Available at: https://www.who.int/medicines/areas/access/OMS_Medicine_prices.pdf.

We used a standard data collection tool (protocol, registration forms) to minimize biases and maximize the collection of valid and reliable data. Local study coordinators undertook regular supervision of the pharmacists and reviewed the adequacy of the data collection.

4) What is the novelty of your work? Several types of studies have been done previously globally? How do you differentiate your work from others?

- As far as we know this is the first study aimed to evaluate patterns of antimicrobial dispensing in community pharmacies in Russia during COVID-19 pandemic. It helps to understand the main reasons why patients visit pharmacies for AM agents purchase. This also helps to assess the real-life practice of antimicrobial choice in the community and allows to make some comparisons with the national guidelines. Since the systems for organizing the provision of medical care differ in different countries, and Russia had its own national clinical guidelines, there are limitations in extrapolating data obtained by other researchers.

5) Literature review and search not proper. Many relevant references are missing. 

- We’ve added a few References.

Reviewer 5 Report

The paper as written should be accepted with minor modification.  This paper provides useful information on dispensing of antibiotics during the COVID pandemic.  

My criticism is some of the English. Examples:  

Line 56/ 57 - ...have been proposed as a part of treatment algorithm...   change to ...have been proposed as a part of the treatment algorithm...  

Line 61 - ...viral infections such as common cold...   change to  ...viral infections such as the common cold...  

Line 101 -   Data was sought from...  change to Data were sought from...  

There are several of these minor English errors.   I did not identify all of them.   I suggest a native English writer read over the manuscript.  

Overall, the paper is acceptable for publication.

Author Response

Reviewer 5

The paper as written should be accepted with minor modification.  This paper provides useful information on dispensing of antibiotics during the COVID pandemic.

- Thank you very much for your review.

My criticism is some of the English. Examples:  

Line 56/ 57 - ...have been proposed as a part of treatment algorithm...   change to ...have been proposed as a part of the treatment algorithm...  
- Corrected

Line 61 - ...viral infections such as common cold...   change to  ...viral infections such as the common cold...  
- Corrected

Line 101 -   Data was sought from...  change to Data were sought from...  
- Corrected

There are several of these minor English errors.   I did not identify all of them.   I suggest a native English writer read over the manuscript.

- Thank you for pointing out these errors. We have made corrections as suggested and double-checked the revised manuscript.

Overall, the paper is acceptable for publication.